# Numerical Study of Disordered Noninteracting Chains Coupled to a Local Lindblad Bath

Viktor Berger[1,2], Andrea Nava[3], Jens H. Bardarson[1], and Claudia Artiaco[1*]

**1** Department of Physics, KTH Royal Institute of Technology, Stockholm 106 91, Sweden
**2** Department of Physics, University of Massachusetts, Amherst, MA 01003, USA
**3** Institut für Theoretische Physik, Heinrich-Heine-Universität, D-40225 Düsseldorf, Germany

* artiaco@kth.se

## Abstract

Disorder can prevent many-body quantum systems from reaching thermal equilibrium, leading to a many-body localized phase. Recent works suggest that nonperturbative effects caused by rare regions of low disorder may destabilize the localized phase. However, numerical simulations of interacting systems are generically possible only for small system sizes, where finite-size effects might dominate. Here we perform a numerical investigation of noninteracting disordered spin chains coupled to a local Lindblad bath at the boundary. Our results reveal strong finite-size effects in the Lindbladian gap in both bath-coupled Anderson and Aubry-André-Harper models, leading to a non-monotonic behavior with the system size. We discuss the relaxation properties of a simple toy model coupled to local Lindblad baths, connecting its features to those of noninteracting localized chains. We comment on the implications of our findings for many-body systems.

# 1   Introduction

In 1958, P. W. Anderson predicted the absence of diffusion in a disordered medium at sufficiently low densities when interactions are negligible [1]. It is now well established that, for random uncorrelated disorder, in $d = 1$ (with $d$ the system spatial dimension) all states are localized for arbitrarily weak disorder; in $d \geq 2$, a metal-insulator transition can happen at finite disorder strength depending on the symmetry class of the model [2–4]. Models with correlated disorder can present a metal-insulator transition even in $d = 1$. This is the case of the Aubry-André-Harper model where disorder arises from the superposition of two lattice potentials with incommensurate wavelengths [5,6].

The critical question arises: does localization persist in the presence of weak interactions? Isolated generic many-body systems are expected to act as thermal baths for themselves, leading to the rapid convergence of local observables to Gibbs ensemble statistics through classical [7–9] or quantum [10,11] chaos. Many-body localization [12,13] has been observed in numerical simulations and experiments across various models (see Ref. 14 for a recent review). Understanding the phenomenology and stability of many-body localized systems is fundamental for advancing our knowledge of out-of-equilibrium phenomena and exploring the limits of statistical mechanics. Additionally, many-body localized systems, thanks to their long-term information storage capability and resilience to noise and dissipation [15], hold promise as quantum memory devices [16,17].

The existence of the many-body localized phase (that is, the persistence of localization in infinitely extended systems at infinite time) is still under debate. For random uncorrelated disorder, it is generally believed that the many-body localization transition cannot occur at finite disorder strengths in $d \geq 2$ [18]. In $d = 1$, its existence has been established for short-range models via perturbation theory analysis within some reasonable assumptions [19–24]. The many-body localized insulator is usually described as an emergent phase of interacting quasilocal integrals of motion, often referred to as $\ell$-bits [25–27]. The $\ell$-bit model has proven to be a valuable tool to explore and understand the phenomenology of many-body localized systems [28–31]. However, recent numerical studies on one-dimensional finite-size systems have questioned the existence of the many-body localized phase at finite disorder strengths [32,33]. For instance, numerical observations of extremely slow particle transport via the number entropy have been interpreted as evidence for the absence of many-body localization [34–37], though this remains debated [38–42]. Other works have revised the many-body localization phase diagram to include the presence of an extended nonlocalized regime, shifting the transition to significantly higher disorder strengths than previously believed [43–47]. Ergodicity-breaking phenomena have also been observed in many-body systems with correlated disorder [48–52], such as in the Aubry-André-Harper model with quasiperiodic onsite potential [53–57]. A complete understanding of the features and stability of many-body localization in this case is similarly lacking. However, recent numerical studies indicate that quasiperiodic systems are less affected by finite-size and finite-time effects than those with uncorrelated dis-

order, suggesting that evidence for the existence of a many-body localized phase may be easier to detect [58, 59]. This might partly be because correlated disorder inhibits the formation of thermal rare regions that can destabilize localization.

Thermal rare regions in uncorrelated disordered systems and the nonperturbative effects they may induce have been the subject of intense research in the last years [43, 44, 60–68]. These rare regions, often termed ergodic bubbles, are expected to arise in parts of the chain where the random potential is unusually uniform, resulting in an effective disorder strength within those parts smaller than in the rest of the chain. Thus, rare regions may thermalize and potentially propagate thermalization throughout the entire system. The so-called avalanche instability argument describes this propagation process via thermal avalanches, identifying the critical disorder strength of the many-body localization transition as the value above which the avalanche halts before thermalizing the full system. The "avalanche instability" is evaluated by comparing Hamiltonian matrix elements to the level spacing of the expanding thermal bubble [14].

In this work, we investigate the avalanche instability in noninteracting disordered spin chains by means of a local Lindblad bath [69–71] acting on the leftmost site, which models the presence of an ergodic bubble at the left boundary. Due to the exponential growth of the Hilbert space dimension with the number of sites, numerical investigations within this framework for interacting chains are restricted to small systems of $\approx$ 10-15 sites [43, 44, 72, 73]. By contrast, the noninteracting case can be studied at the single-particle level, allowing us to examine systems up to 120 sites and perform finite-size scaling analyses. We focus on the Anderson and Aubry-André-Harper chains, both of which exhibit a localized phase in closed setups. We analyze the behavior of the Lindbladian spectral gap, the inverse of which corresponds to the slowest relaxation time of the chain, as we vary system size and disorder strength. Our findings reveal the presence of strong finite-size effects. The behavior of the Lindbladian gap for small system sizes suggests that the critical disorder for the avalanche instability increases with system size. However, at larger system sizes, both noninteracting chains exhibit signatures of localization for progressively lower disorder strengths, making the avalanche critical disorder drift to smaller values. Similar finite-size effects could influence results within the local Lindblad bath framework for interacting disordered chains. We also study the overlap of single-particle eigenstates with the leftmost site of the chain as a function of system size and disorder strength. Finally, we investigate the relaxation properties of a spinless-fermion toy model coupled with local Lindblad baths, connecting its features to those of noninteracting localized chains when the number of sites is sent to infinity first.

## 2 Models and methods

### 2.1 Disordered noninteracting chains

We consider tight-binding models of spinless fermions on a chain with nearest-neighbor hopping and randomly distributed onsite potential under open boundary conditions. Cast in a second-quantized form, the Hamiltonian reads

$$H = \frac{J}{2} \sum_{i=1}^{L-1} (c_i^\dagger c_{i+1} + \text{H.c.}) + \sum_{i=1}^{L} h_i c_i^\dagger c_i = \sum_{i,j=1}^{L} c_i^\dagger \mathcal{H}_{ij} c_j, \tag{1}$$

where $c_i^\dagger$ ($c_i$) are the creation (annihilation) spinless fermionic operators satisfying the canonical anticommutation relations $\{c_i, c_j^\dagger\} = \delta_{i,j}$ with $\delta_{i,j}$ the Kronecker delta. The $L \times L$ Hermitian tridiagonal matrix $\mathcal{H}$ is the single particle Hamiltonian with matrix elements $\mathcal{H}_{i,j} =$

$\frac{J}{2}(\delta_{i,j+1} + \delta_{i+1,j}) + h_i \delta_{i,j}$, where $J$ is the hopping strength and $h_i$ are the onsite energies. The Hamiltonian (1) conserves the particle number $N = \sum_i n_i$ with $n_i = c_i^\dagger c_i$.

Since the model is noninteracting, we can diagonalize the Hamiltonian (1) by means of the operators

$$f_\alpha^\dagger = \sum_{i=1}^{L} \Psi_i^\alpha c_i^\dagger, \tag{2}$$

and their Hermitian conjugates $f_\alpha$. Here $\Psi^\alpha$ are the eigenfunctions associated with the eigenenergies $\epsilon_\alpha$ of the single particle Hamiltonian $\mathcal{H}$. Then,

$$H = \sum_{\alpha=1}^{L} \epsilon_\alpha f_\alpha^\dagger f_\alpha. \tag{3}$$

Hence, the computational cost of the problem scales linearly with system size.

By means of the Jordan-Wigner transformation, the Hamiltonian (1) can be recast as the XX-spin-chain Hamiltonian with an onsite field along $\hat{z}$:

$$H = J \sum_{i=1}^{L-1} \left( S_i^x S_{i+1}^x + S_i^y S_{i+1}^y \right) + \sum_{i=1}^{L} h_i S_i^z, \tag{4}$$

where $S_i^\alpha = \frac{1}{2}\sigma_i^\alpha$ with $\sigma_i^\alpha$ for $\alpha = x, y, z$ the Pauli matrices.

### 2.1.1 The Anderson chain

In the Anderson model, the onsite potential $h_i$'s are independent identically distributed random variables. We take $h_i$ to be drawn from a uniform probability distribution:

$$h_i \in [-W, W], \tag{5}$$

with average $\bar{h} = 0$ and standard deviation $\sigma_h^2 = W^2/3$. $W$ characterizes the disorder strength. For $W = W_c = 0$, the single-particle eigenenergies are $\epsilon_\alpha = 2J\cos(k_\alpha)$ (assuming unit lattice spacing), with $k_\alpha = \pi\alpha/(L+1)$ and $\alpha$ integer, and the associated eigenfunction coefficients are $\Psi_i^\alpha = \mathcal{N}\sin(k_\alpha i)$ with $\mathcal{N}$ a normalization constant. For $W > W_c = 0$ the eigenfunctions $\Psi^\alpha$ have an exponentially decaying envelope in space,

$$|\Psi_i^\alpha| \sim e^{-|i - n_\alpha|/\xi_\alpha}, \tag{6}$$

with $\xi_\alpha$ a characteristic decay length, known as the localization length, and $n_\alpha$ the localization center [4]. The localization length $\xi_\alpha$ depends on both the disorder strength and the eigenenergy $\epsilon_\alpha$. The energy-averaged localization length diverges at weak disorder as $\sim 1/W^2$ [74,75], while at strong disorder it goes like $\sim 1/(2\ln(W))$ [76].

Let us note that finite-size effects already appear at this level. If in a finite chain of length $L$ typically $\xi_\alpha \gg L$ (weak disorder) then the system eigenfunctions generally extend across the entire chain, with a nonzero overlap on both edges. However, the insulating behavior is always recovered in the thermodynamic limit for any finite disorder strength $W$. It follows that, for weak disorder, a proper finite-size scaling analysis should be performed considering system sizes larger than the localization length.

### 2.1.2 The Aubry-André-Harper chain

In the Aubry-André-Harper model [5,6] $h_i$ is a quasi-periodic function:

$$h_i = \lambda \cos(2\pi\beta i + \varphi), \tag{7}$$

where $\lambda$ is the disorder strength, $\beta$ is a period incommensurate to the lattice period and $\varphi$ is a phase. In this work we set $\beta$ to be the golden ratio $(\sqrt{5}-1)/2$. To emulate the presence of disorder the phase $\varphi$ is drawn from a uniform probability distribution

$$\varphi \in [0, 2\pi) \tag{8}$$

The Aubry-André-Harper model possesses a self-duality which elucidates its localization properties. By performing the transformation

$$c_n^\dagger = e^{i\varphi n} \sum_{m=1}^{L} e^{im(2\pi\beta n + \varphi)} \tilde{c}_m^\dagger \tag{9}$$

where $\tilde{c}_i$ and $\tilde{c}_i^\dagger$ are fermionic operators, the Aubry-André-Harper Hamiltonian is mapped to

$$H = \frac{\lambda}{2} \sum_{i=1}^{L} (\tilde{c}_i^\dagger \tilde{c}_{i+1} + \text{H.c.}) + J \sum_{i=1}^{L} \cos(2\pi\beta i + \varphi) \tilde{c}_i^\dagger \tilde{c}_i. \tag{10}$$

The Hamiltonian (10) corresponds to the original Hamiltonian with the substitution $J \to \lambda$ and $\lambda \to J$. By the Heisenberg uncertainty principle, when one of the two Hamiltonians has localized eigenstates, the other one must have extended eigenstates [77]. We can then conclude that there must be a localization transition when the model maps to itself, that is, when $\lambda = \lambda_c = J$. For disorder values $\lambda > \lambda_c$ the single-particle eigenfunctions have exponentially decaying envelopes, as discussed for the Anderson model in Eq. (6). By means of the Thouless formula [78], the localization length is

$$\xi^{-1} = \log \frac{\lambda}{J}; \tag{11}$$

In the Aubry-André-Harper model, $\xi$ is independent of the eigenstate eigenenergy [6].

## 2.2 The avalanche instability argument

### 2.2.1 Original formulation

The avalanche instability argument [14, 60] describes a possible nonperturbative mechanism for the many-body localization transition. It posits that ergodic bubbles in disordered many-body systems can destabilize many-body localization by initiating a thermal avalanche that spreads throughout the entire system. In a one-dimensional chain with uncorrelated disorder, rare spatial regions exist where the random terms (for instance, random onsite potentials) are unusually uniform. These regions are therefore characterized by an effective disorder strength smaller than the rest of the system and form local ergodic bubbles. The avalanche theory assumes that an ergodic bubble of $N$ spins tends to expand by thermalizing localized surrounding spins that then become part of the ergodic bubble. The typical rate at which localized spins located at distance $\ell$ from the bubble become thermal is exponentially small in $\ell$ and goes as $\sim \kappa^{-\ell}$, where $\kappa$ is a disorder-dependent parameter that increases as disorder increases. Assuming the ergodic bubble initially resides in the bulk of the chain, after the avalanche has propagated a distance $\ell$ in both left and right directions, the thermalized region includes $N + 2\ell$ spins and its many-body level spacing is $\sim 2^{-(N+2\ell)}$. According to the avalanche theory, the thermal bubble can act as a reservoir for surrounding localized spins if the spin thermalization rate surpasses the level spacing of the thermal bubble as, under this condition, the spectrum of the bubble appears "continuous" to the spin. Hence, for $\kappa < 4$ the avalanche proceeds and slowly thermalizes the entire chain, whereas for $\kappa \geq 4$ the avalanche halts and localization is stable. The growth of the bubble ceases when $\kappa^{-\ell} \sim 2^{-(N+2\ell)}$. Thus, according to the previous

arguments, $\kappa \simeq 4$ sets the critical disorder strength for the many-body localization transition. Notice that the enlargement process of the ergodic bubble by thermalizing neighboring spins is highly nonperturbative. The avalanche theory predicts the absence of the many-body localization transition at finite disorder strength in dimensions larger than one. The relevance of the avalanche mechanism for one-dimensional systems remains debated, as numerical studies have not provided conclusive results yet [60, 66].

Thermal avalanches have been numerically investigated within diverse frameworks, for instance, by modeling the ergodic bubble by a random matrix [61, 63], via a planted thermal inclusion [64, 68], or as a local Lindblad bath acting at the chain boundary [43, 44, 72, 73]. In the last approach, the avalanche instability is probed by computing the spectral gap of the Lindblad superoperator, denoted as $\Delta$, as an estimate for the smallest relaxation rate of the system (see Sec. 2.2.2 below). By assuming that the avalanche mechanism takes place and that $\Delta$ obtained within the Lindblad framework scales as the spin thermalization rate induced by an actual thermal avalanche seeded in the bulk of the chain, the avalanche critical disorder corresponds to $\Delta \sim 4^{-L}$. Thus, if $\Delta$ decreases with system size $L$ slower than $4^{-L}$ then the avalanche is capable of thermalizing the entire chain, whereas if $\Delta$ decays faster than $4^{-L}$ then localization is stable. By the behavior of $\Delta 4^L$ for different system sizes and disorder strengths for systems of up to 15 sites, the critical disorder for the many-body localization transition in the XXZ spin chain was estimated to be significantly higher than suggested by earlier numerical studies [43, 44]. Later works have however pointed out that observing the termination of quantum avalanches seeded by ergodic bubbles requires preparing the system in an eigenstate of the localized Hamiltonian [68]. Under this condition, the avalanche critical disorder aligns with the critical value obtained from exact diagonalization studies [79]. Assumptions in the local Lindblad bath framework about ergodic bubble properties may therefore overestimate the critical disorder strength.

To shed further light on the avalanche instability argument within the local Lindblad bath framework, in this article we apply it to noninteracting spin (equivalently, spinless-fermion) disordered chains. We assume that the local Lindblad bath acting at the left boundary of the chain provides an estimate of the chain smallest relaxation rate when an ergodic bubble is initially located at the left boundary. The bubble can therefore spread only towards the right side. Hence, $\kappa \simeq 2$ sets the avalanche critical disorder strength. Notice that both ergodic bubbles and thermal avalanches can arise only in the presence of interactions. In particular, the onset of avalanches relies on the fact that spins can influence their neighbors' thermalization process as they become part of the ergodic bubble. This cannot happen in noninteracting localized chains where single-particle eigenfunctions are localized in real space and transport is suppressed. Nevertheless, since the local Lindblad bath framework a priori assumes that the avalanche takes place, the scaling of the Lindbladian gap $\Delta$ still provides an estimate of the avalanche critical disorder strength. Such a value is larger than the actual critical disorder of the closed chains ($W_c$ and $\lambda_c$ defined in Sec. 2.1) and is set by the localization length of the Lindbladian gap $\zeta$, where $\Delta \propto e^{-L/\zeta}$ [80]. Noninteracting localized chains can be seen as a proxy for the interacting model deep in the localized phase.

### 2.2.2 Modeling avalanches via the Lindblad master equation

We model thermal avalanches in noninteracting disordered chains via a local Lindblad bath coupled to the chain leftmost site where the ergodic bubble is assumed to reside. The dynamics of the spin chain is described by the Lindblad master equation [69–71]:

$$\frac{d\rho}{dt} = \mathcal{L}[\rho],$$ (12)

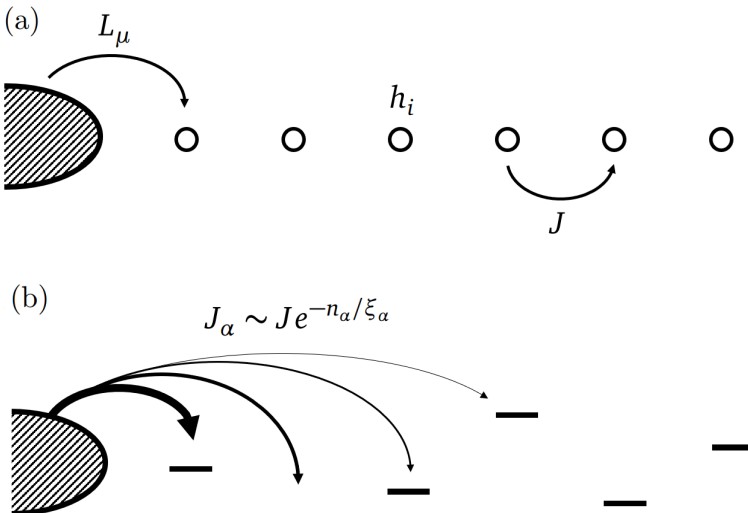

Figure 1: (a) Sketch of the local Lindblad bath framework in Eqs. (13)-(15). The chain (black circles) is characterized by the Hamiltonian $H$ that can be either the Anderson or the Aubry-André-Harper one [Eqs. (4), (5), (7)]. The bath on the left is coupled via $L_1$ and $L_2$ to the leftmost site of the chain. (b) Effect of the bath on localized noninteracting spin chains. The Lindblad jump operators give rise to hoppings from the bath to the single-particle Hamiltonian eigenstates labeled by $\alpha$, with strength $J_\alpha$ exponentially decaying with the localization center $n_\alpha$, as exemplified by the shading arrows. The onsite bars at different heights represent the random single-particle eigenenergies $\epsilon_\alpha$.

with

$$\mathcal{L}[\rho] = -i[H, \rho] + \sum_{\mu} \gamma_{\mu}\left(L_{\mu}\rho L_{\mu}^{\dagger} - \frac{1}{2}\{L_{\mu}^{\dagger}L_{\mu}, \rho\}\right), \tag{13}$$

where $[\cdot, \cdot]$ is the commutator, $\{\cdot, \cdot\}$ the anticommutator, and $\gamma_{\mu} \geq 0$ are the damping rates. The Planck constant $\hbar$ is set to unity throughout the text. The first term on the right-hand side of Eq. (13) corresponds to the von Neumann equation and gives the coherent evolution of the system density matrix under its own Hamiltonian $H$. For simplicity, we neglect the Lamb shift of the system Hamiltonian [81]. The second term accounts for incoherent processes (that is, dephasing and dissipation) generated by the coupling to the bath.

We assume that the bath is able to flip the spin on the first site of the chain. This process is modeled by the Lindblad jump operators

$$L_1 = S_1^+, \ L_2 = S_1^-, \tag{14}$$

which are the raising and lowering operators acting on the leftmost site, defined as $S_1^{\pm} = S_1^x \pm i S_1^y$. In terms of fermionic coordinates (up to an irrelevant phase factor), these processes correspond to locally creating or annihilating fermions from the leftmost site of the chain [81], that is

$$L_1 = c_1^{\dagger}, \ L_2 = c_1. \tag{15}$$

In addition, we assume that the bath is at infinite temperature: $\gamma_1 = \gamma_2 = \gamma$. Eq. (13), together with the jump operators in Eqs. (14)-(15), has been widely employed to describe out-of-equilibrium dynamics and relaxation in both interacting and noninteracting spin and fermionic chains reproducing physically sound results [82–87].

The solution to the Lindblad equation can be written as

$$\rho(t) = \sum_{i=1}^{N} \sum_{m=1}^{M_i} a_{i,m} e^{\lambda_i t} R_{i,m} \tag{16}$$

where $\lambda_i$ are $N$ distinct eigenvalues of $\mathcal{L}$ with multiplicity $M_i$, $R_{i,m}$ are the corresponding right eigenmatrices, and $a_{i,m} = \text{Tr}(E_{i,m}\rho(0))$ are the overlaps of the initial density matrix with the left eigenmatrices $E_{i,m}$ of $\mathcal{L}$. There exists at least one right eigenmatrix with eigenvalue $\lambda_0 = 0$ corresponding to a stationary solution. For all eigenvalues $\lambda_i$ it holds that $\text{Re}(\lambda_i) \leq 0$. The quantity $-\text{Re}(\lambda_1)$, with $\lambda_1$ the eigenvalue with the largest real part smaller than zero, can be interpreted as the relaxation rate of the system $\Delta$ [88].

The bath-system model is sketched in Fig. 1(a). The Lindblad jump operators $L_1$ and $L_2$ can be expanded in terms of the $f_{\alpha}$ and $f_{\alpha}^{\dagger}$ operators in Eq. (2). It follows that there is a direct coupling between the bath and the $\alpha$-th single-particle Hamiltonian eigenstate. The jump operators allow the bath to inject and absorb magnetization (or particles) in the eigenstates. As the Lindblad operators act on the leftmost site of the chain, the coupling strength is proportional to $|\Psi_1^{\alpha}|$. Thus, due to the localization of the eigenfunctions $\Psi^{\alpha}$ in space [see Eq. (6)], the coupling strength decays exponentially with localization center $J_{\alpha} \sim J e^{-n_{\alpha}/\xi_{\alpha}}$, as depicted in Fig. 1(b). The bath only significantly affects eigenstates that have a nonvanishing overlap with the leftmost site of the chain. If the number of sites in the chain is sent to infinity *first*, an infinite number of eigenstates becomes effectively decoupled from the bath. Therefore, within the framework in Eqs. (13)-(15), regardless of the presence of the bath, an infinitely extended noninteracting localized system remains localized.

Among the Lindblad jump operators, one might in principle also include a local dephasing term, $L_3 = S_1^z$ in spin coordinates or $L_3 = 2c_1^{\dagger}c_1$ in fermionic ones. While in general dephasing terms appropriately distributed along the bulk of the chain can induce sub-diffusive/diffusive transport in noninteracting localized systems [89–92], in App. B we show that if the dephasing acts locally only at the system boundary it does not qualitatively affect the results.

We compute $\Delta$ by means of third quantization [93]. Third quantization is a general method to solve the Lindblad master equation for noninteracting spinless fermionic systems, provided that the Lindblad jump operators are linear in the fermionic variables, as verified by Eqs. (14)-(15). The resources required by this method scale linearly with the system size, unlike the exponential scaling needed to solve the Lindblad master equation (13) directly.

## 2.3 Observables

We compute the Lindbladian gap $\Delta = -\text{Re}(\lambda_1)$. In the localized phase, $\Delta$ is exponentially suppressed with system size, while in the ergodic phase it decays as a power law governed by hydrodynamic modes [80, 88, 93]. We investigate the behavior of $\Delta 2^L$ as a function of system size and disorder strength. As $2^L$ is the number of Hamiltonian eigenstates in a spin chain of size $L$, its comparison with $\Delta$ can be interpreted as a measure of the fraction of eigenstates with non-negligible overlap with the leftmost site of the chain to which the bath is coupled. We attribute an increase of $\Delta 2^L$ with $L$ to the fact that the finite-size chain is in the delocalized regime, while if $\Delta 2^L$ decreases with $L$ it is in the localized one. Thus, $\Delta \sim 2^L$ sets a system-size dependent avalanche critical disorder strength for both the Anderson and Aubry-André-Harper models. As discussed in App. A.2, scaling $\Delta$ by $4^L$ as done in the interacting case [43, 44] does not alter the qualitative features observed in our numerical results. It however provides a larger value of the avalanche critical disorder strength than $2^L$.

Additionally, we compute the overlap of the single-particle eigenstates with the leftmost site of the chain, $p$. Specifically, we diagonalize the single-particle Hamiltonian, compute the overlap of the eigenstates on the leftmost physical site, and define $p$ as the percentage of overlaps greater than $10^{-14}$.

# 3 Numerical results

## 3.1 The Lindbladian gap

Fig. 2 illustrates the behavior of the disorder-averaged rescaled Lindbladian gap $\langle \log_{10}(\Delta 2^L) \rangle$ for both the bath-coupled Anderson [(a)-(b)] and Aubry-André-Harper [(c)-(d)] chains as system size and disorder strength are varied. Let us recall that we interpret an increase of $\langle \log_{10}(\Delta 2^L) \rangle$ with system size as a signature of ergodicity while its decrease as a signature of localization. Panel (a) shows that the dependence of $\langle \log_{10}(\Delta 2^L) \rangle$ on $L$ in the Anderson case is highly nonmonotonic, indicating first a localized (inset), then an ergodic, and finally again a localized regime as $L$ increases up to 120 for disorder strengths $W \gtrsim 0.25$. Indeed, for such disorder strengths, $\langle \log_{10}(\Delta 2^L) \rangle$ increases at intermediate scales and then curves downwards at a threshold scale $L^*$ that increases with decreasing disorder. This intermediate-scale ergodic regime is caused by the presence of strong finite-size effects, as already observed in a similar context [94]. For the system sizes accessible to us, it is hard to predict whether smaller disorder values, such as $W = 0.2$, will show localization features at system sizes larger than the ones accessible to us. Panel (c) depicts $\langle \log_{10}(\Delta 2^L) \rangle$ vs. $L$ for the Aubry-André-Harper model. Here, finite-size effects are weaker than in the Anderson case and the slopes of the curves do not change significantly up to the largest system sizes reachable to us. For instance, at $\lambda = 1.3$, $\langle \log_{10}(\Delta 2^L) \rangle$ shows a small increase at small system sizes and then linearly decreases for larger values of $L$.

To estimate the avalanche critical disorder $W^*$ ($\lambda^*$), we employ linear interpolation to compute the intersection point between curves of $\langle \log_{10}(\Delta 2^L) \rangle$ as a function of $W$ ($\lambda$) for different values of $L$, as reported in panels (b) and (d). We consider the intersection for close $L$ values (for instance, $L = 6$ with $L = 4$, $L = 8$ with $L = 6$, etc.) as an estimate of $W^*(L)$

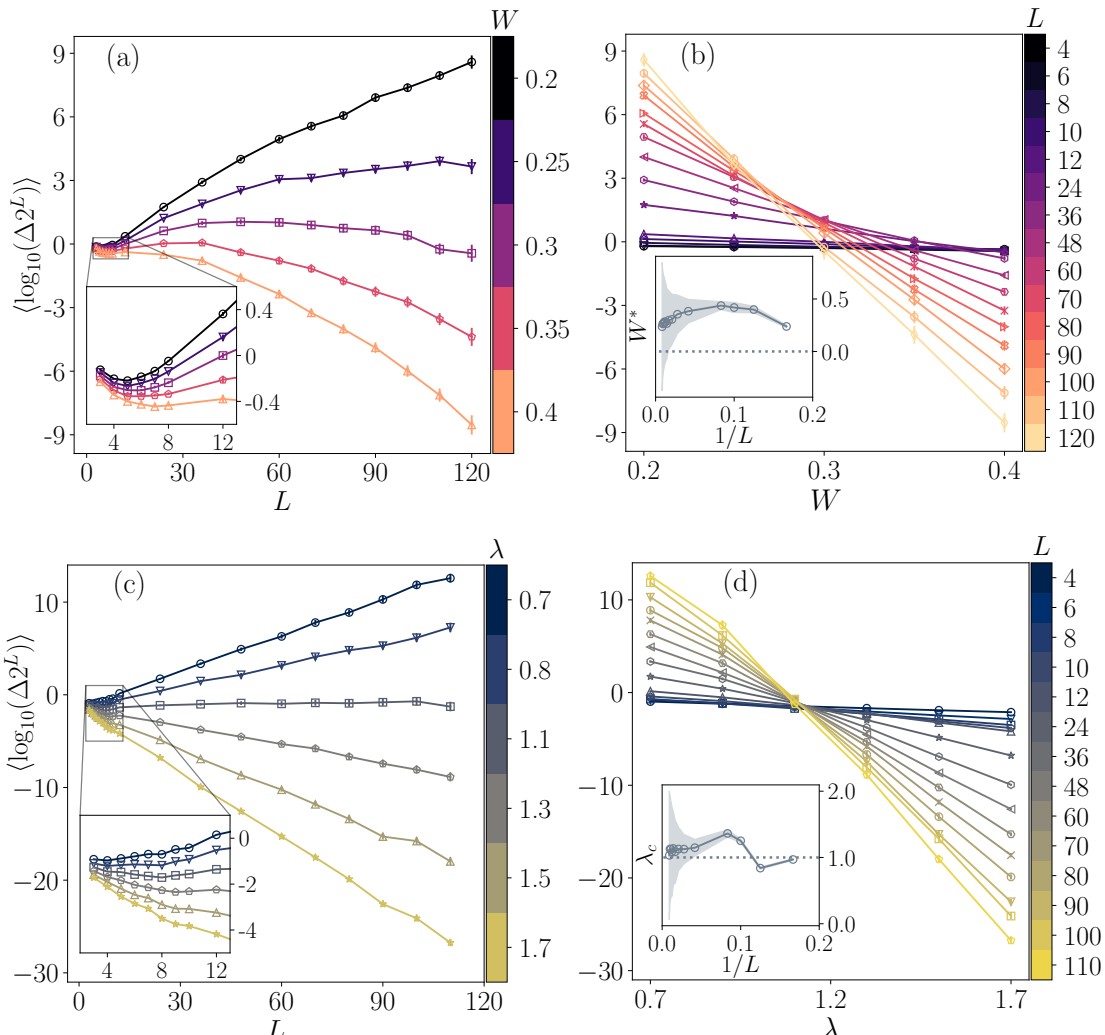

Figure 2: $\langle \log_{10}(\Delta 2^L) \rangle$ for the Anderson [(a)-(b)] and the Aubry-André-Harper [(c)-(d)] chains coupled to a local Lindblad bath acting on the leftmost site as a function of system size $L$ and disorder strength $W$ or $\lambda$. Dotted lines in the insets of (b) and (d) mark the critical disorder strength of the closed chains: $W_c = 0$ for the Anderson chain and $\lambda_c = 1$ for the Aubry-André-Harper chain. We set $J = \gamma = 1$ and average over at least 2000 (1000) disorder realizations for the Anderson (Aubry-André-Harper) model for each data point. Error bars are 95% bootstrap confidence intervals.

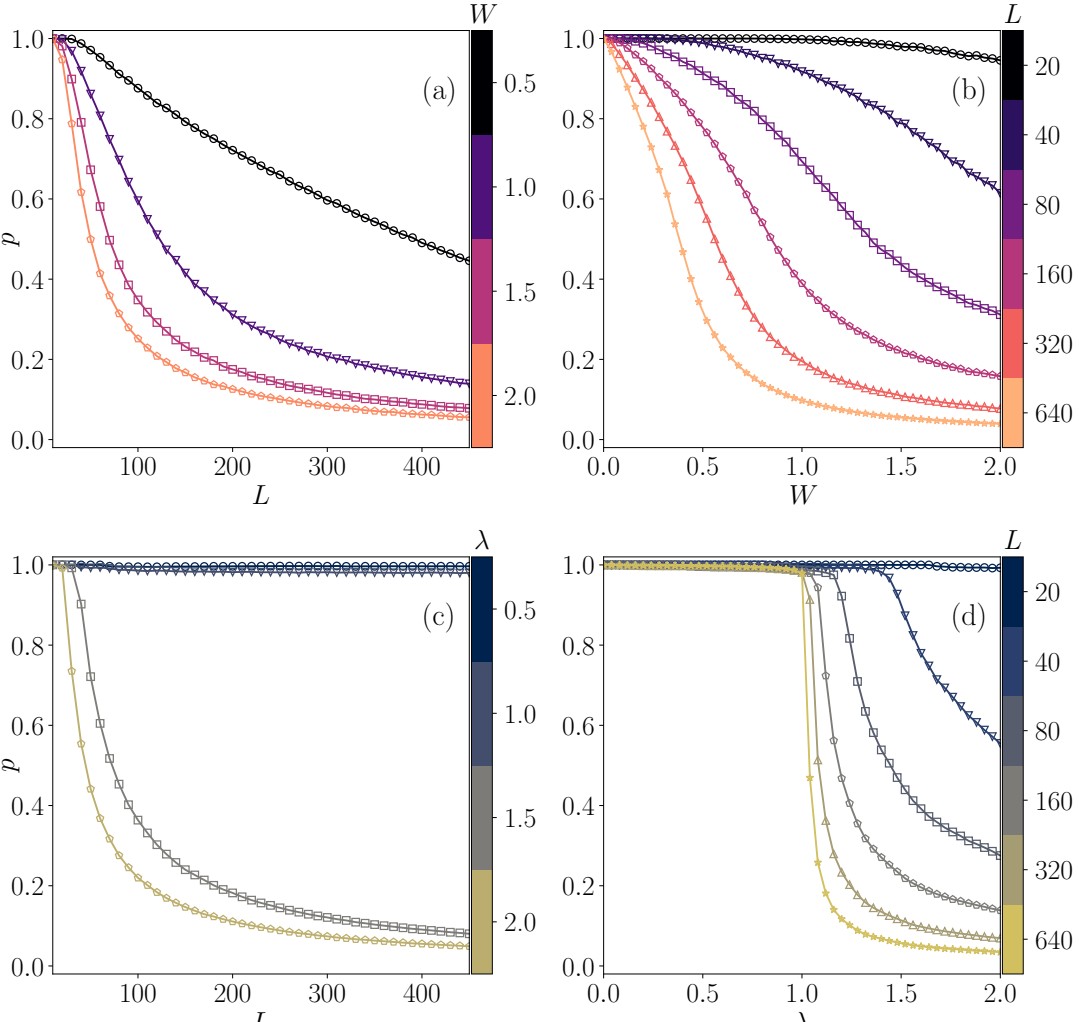

Figure 3: Percentage of single-particle Hamiltonian eigenstates with nonvanishing (that is, larger than $10^{-14}$) overlap on the leftmost physical site $p$ of the Anderson and Aubry-André-Harper chains as a function of system size [(a) and (c)] and disorder strength [(b) and (d)]. (a)-(b) For the Anderson chain, $p$ rapidly decreases as $L$ and $W$ increase. (c)-(d) For the Aubry-André-Harper chain, $p \approx 1$ for $\lambda \lesssim \lambda_c = 1$, whereas it rapidly decreases for $\lambda \gtrsim \lambda_c$. We set $J = 1$ and average over 100 disorder realizations for each data point. Error bars are nearly invisible.

($\lambda^*(L)$). Results are shown in the insets, where the dotted lines mark the critical disorder of the closed chains: $W_c = 0$ and $\lambda_c = 1$. Inset of (b) illustrates that $W^*(L)$ increases with $L$ at small $L$, while for larger system sizes it has a significant drift towards lower values. A faithful extrapolation of the critical disorder in the limit $L \to \infty$ from our finite-size results is difficult. A quadratic fit of $W^*$ vs. $1/L$ for system sizes $L = 24, 36, \ldots, 120$ gives the estimate $W^*_\infty \approx 0.2$. The extrapolated $W^*_\infty$ further decreases as smaller systems are excluded from the fit. This again highlights the presence of strong finite-size effects. Inset of (d) shows that for small system sizes $\lambda^*$ increases with $L$, while already at intermediate $L$ values the critical disorder hovers around $\lambda^* \approx 1.1$, confirming that the Aubry-André-Harper model is less affected by finite-size effects. A quadratic fit of $\lambda^*$ vs. $1/L$ for system sizes $24, 36, \ldots 120$ gives $\lambda^*_\infty \approx 1.05$.

Our findings highlight that an estimate of the avalanche critical disorder for noninteracting chains within the local Lindblad bath framework is subject to strong finite-size effects suggesting a higher critical value. Similarly strong finite-size effects might be present for interacting disorder chains. Without a definitive scaling theory for the many-body localization transition, interpreting finite-size numerics even within the local Lindblad bath framework appears challenging [95, 96].

## 3.2 Overlap of single-particle eigenstates on the leftmost site

Fig. 3 shows that, as system size or disorder increases, the percentage $p$ of single-particle Hamiltonian eigenstates with nonvanishing (that is, larger than $10^{-14}$) overlap on the leftmost site rapidly decreases. Since the bath-system coupling is exponentially small in the distance between the bath and the localization center of the eigenstates $n_\alpha$ [see Fig. 1(b)], the bath is less effective in creating or annihilating particles in eigenstates with vanishing overlap $p$. We observe the presence of strong finite-size effects in the Anderson model in panels (a)-(b): for $L = 20$, $p \approx 1$ for $W \leq 2$; for $W = 0.5$, $p \approx 1$ for $L \leq 40$. In the Aubry-André-Harper model, at $\lambda < \lambda_c = 1$, $p$ remains almost constant and close to 1 as expected for a delocalized system. However, as the disorder strength increases above the critical value $\lambda > 1$, $p$ rapidly vanishes for sufficiently large system sizes [panel (c)]. Finite-size effects are present, such as $p \approx 1$ for $L = 20$ and $\lambda \lesssim 2$ [panel (d)] but weaker than in the Anderson case. This is clearly visible by comparing, for instance, $p$ vs. $W$ and $\lambda$ at $L = 640$ in panels (a) and (c).

Notice that, if one takes the thermodynamic limit first, in noninteracting localized chains at any disorder strength an infinitely large number of eigenstates becomes decoupled from the bath and the local Lindblad bath framework is unable to induce transport (equivalently, the system relaxation time goes to infinity). A toy model describing the physics of an infinitely extended noninteracting localized chain coupled to a local Lindblad bath is discussed in Sec. 4.

# 4 A toy model for local baths and relaxation

In the previous sections, we observed that, in disordered noninteracting chains coupled to a local Lindblad bath, single-particle localization leads to an exponential growth of the relaxation time $\Delta^{-1}$ as a function of $L$. The relaxation time is linked to the overlap of single-particle eigenstates on the site(s) where the bath is coupled. In both the Anderson and Aubry-André-Harper chains, the eigenstate weight on the leftmost site of the chain—where the bath is connected—approaches zero as the localization center increases or the localization length decreases (Fig. 3). By sending the number of sites of the chain to infinity first, the number of eigenstates with finite overlap on the leftmost site becomes negligible, effectively prohibiting transport even in the presence of the bath. In this section, we examine a simple toy model to capture possible relaxation behaviors in spinless fermionic systems coupled to local baths.

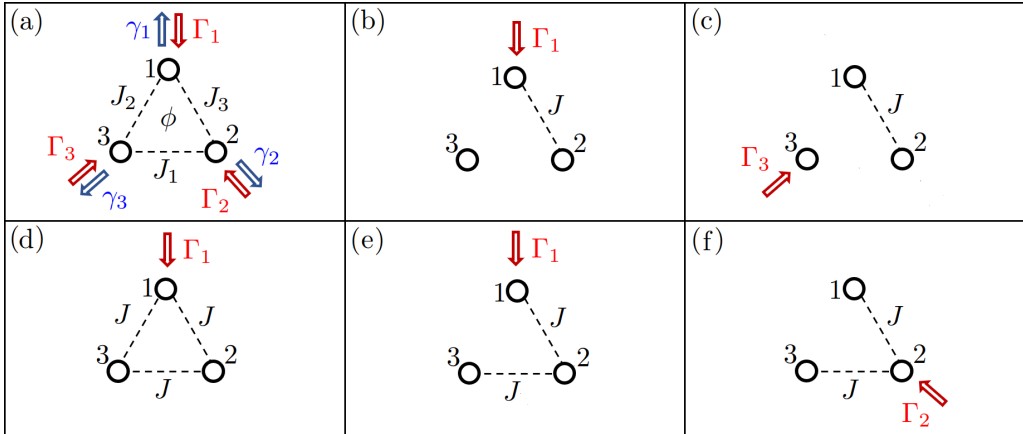

Figure 4: (a) A three-site ring (trimer) pierced by a magnetic flux and coupled to three different local baths creating and annihilating spinless fermions on each site. (b) One site is spatially decoupled from the other two that form a dimer; a local bath acts on one of the sites of the dimer. (c) Same as (b) but with the bath acting on the decoupled site. (d) A trimer with periodic boundary conditions in which the bath acts only on a single site. (e) A trimer with open boundary conditions coupled to a bath acting on one boundary site. (f) Same as (e) but with a local bath acting only on the central site.

Small quantum-dot systems with a reduced number of degrees of freedom are a common experimental and theoretical playground to investigate the thermalization properties of physical systems [97–103]. We consider a three-site ring, referred to as a trimer, coupled to local external baths that create and annihilate spinless fermions on each site. Similar models have been investigated in different contexts [104–107]. Despite its simplicity, this model exhibits some interesting properties useful to show how a system connected to local baths can exhibit a divergent relaxation time when some of its degrees of freedom are decoupled (in real space or in the basis of the Hamiltonian eigenstates) from the site connected to the bath. The Hamiltonian for the three-site ring depicted in Fig.4(a) is

$$H_{\text{trimer}} = -\sum_{i=1}^{3} J_i e^{i\phi/3} c_{i-1}^\dagger c_{i+1} + \text{H.c.},\tag{17}$$

where $J_i$ is the hopping term between site $i \mp 1$ and site $i \pm 1$, with $4 \equiv 1$ and $0 \equiv 3$, and $\phi$ is a magnetic flux piercing the ring. The coupling with the external baths is described in terms of the local Lindblad operators

$$L_i^{(c)} = \sqrt{\Gamma_i} c_i^\dagger \quad , \quad L_i^{(a)} = \sqrt{\mu_i} c_i,\tag{18}$$

that create ($c$) and annihilate ($a$) a particle on site $i$, plus a dephasing ($d$) term

$$L_i^{(d)} = \sqrt{\nu_i} c_i^\dagger c_i.\tag{19}$$

As the Hamiltonian is quadratic in the creation and annihilation operators and the Lindblad jump operators are either linear in them or proportional to the onsite number operator, we can write a closed set of equations for the correlation matrix $\mathcal{C}$, with matrix elements $\mathcal{C}_{i,j} = \langle c_i^\dagger c_j \rangle$, given by [86,91]

$$\frac{d}{dt}\mathcal{C}_{i,j} = i\left[\mathcal{H}_{\text{trimer}}^t, \mathcal{C}\right]_{i,j} - \frac{1}{2}\left\{(\hat{\Gamma} + \hat{\mu}), \mathcal{C}\right\}_{i,j} + \hat{\Gamma}_{i,j} + \frac{\nu_i + \nu_j}{2}\left(\delta_{i,j} - 1\right)\mathcal{C}_{i,j},\tag{20}$$

where $\mathcal{H}_{\mathrm{trimer}}^t$ is the transpose of the single particle Hamiltonian

$$\mathcal{H} = \begin{pmatrix} 0 & -J_3 e^{-i\phi/3} & -J_2 e^{i\phi/3} \\ -J_3 e^{i\phi/3} & 0 & -J_1 e^{-i\phi/3} \\ -J_2 e^{-i\phi/3} & -J_1 e^{i\phi/3} & 0 \end{pmatrix}, \tag{21}$$

such that $H_{\mathrm{trimer}} = c^\dagger \mathcal{H}_{\mathrm{trimer}} c$, while $c = (c_1, c_2, c_3)^t$, $\hat{\Gamma} = \mathrm{diag}\{\Gamma_1, \Gamma_2, \Gamma_3\}$, and $\hat{\mu} = \mathrm{diag}\{\mu_1, \mu_2, \mu_3\}$. The diagonal elements of $\mathcal{C}$ are the expectation values of the number operator on each site, $\mathcal{C}_{i,i} = \langle n_i \rangle$. The off-diagonal elements are instead related to the current across each pair of sites, $\langle I_{i,i+1} \rangle = 2J_{i-1}\mathrm{Im}[e^{-i\phi/3}\mathcal{C}_{i,i+1}]$. For the trimer system we can write

$$\frac{d}{dt}\mathcal{C}_{i,i} = \Gamma_i \left(1 - \mathcal{C}_{i,i}\right) - \mu_i \mathcal{C}_{i,i}$$
$$+ i\left[ J_{i-1}\left(e^{-i\phi/3}\mathcal{C}_{i,i+1} - e^{i\phi/3}\mathcal{C}_{i,i+1}^*\right) - J_{i+1}\left(e^{-i\phi/3}\mathcal{C}_{i-1,i} - e^{i\phi/3}\mathcal{C}_{i-1,i}^*\right)\right] \tag{22}$$

$$\frac{d}{dt}\mathcal{C}_{i,i+1} = -\frac{1}{2}\left(\mu_i + \Gamma_i + \nu_i + \mu_{i+1} + \Gamma_{i+i} + \nu_{i+1}\right)\mathcal{C}_{i,i+1}$$
$$+ i\left[ J_{i-1}e^{i\phi/3}\left(\mathcal{C}_{i,i} - \mathcal{C}_{i+1,i+1}\right) - J_{i+1}e^{-i\phi/3}\mathcal{C}_{i+1,i-1}^* + J_i e^{-i\phi/3}\mathcal{C}_{i-1,i}^*\right]. \tag{23}$$

We now focus on five special cases, shown in panels (b)-(e) in Fig. 4. For simplicity, we assume that in each case the system is coupled to a single local bath that can only create particles. Thus, we set the annihilation and dephasing coefficients to zero, that is, $\mu_i = \nu_i = 0$, and assume that $\Gamma_i$ is nonzero only on a single site of the trimer. Case (b) is the simplest one. It consists of a system of three sites with one of the sites (site 3) totally decoupled from the other two, as well as from the bath, which acts locally on site 1. We are interested in the total occupation number $N = \sum_{i=1}^3 n_i$ at $t \to \infty$. Since site 3 is completely decoupled from the rest of the system, its occupancy is fixed by the initial condition and cannot evolve with time. Indeed, from Eq. (22) it is easy to see that $\langle \dot{n}_3 \rangle = 0$. It follows that for $t \to \infty$ we have an infinite set of solutions in which $\langle n_3 \rangle$ is a free parameter set only by the initial condition and $\langle n_1 \rangle = \langle n_2 \rangle = 1$. At $t \to \infty$ the system reaches a stationary solution which does not necessarily have full occupancy $N = 3$, unless $\langle n_3 \rangle = 1$ from the beginning.

Case (c) is a slight variation of case (b). Site 3 is disconnected from the other sites, but it is coupled to the bath. Hence, the expectation value of the number operator in 3 goes to 1: $\lim_{t\to\infty}\langle n_3 \rangle = 1$. However, since the other two sites are decoupled from the bath, the total number of particles on them is preserved, that is, $\langle \dot{n}_1 \rangle + \langle \dot{n}_2 \rangle = 0$. If the dimer composed by sites 1 and 2 is not characterized by a thermal density matrix at time $t = 0$, it will never be in a thermal equilibrium state, and an oscillating behavior of the occupancies of sites 1 and 2 will persist at any time. As in case (b), the system is not able to reach the full occupancy $N = 3$ at $t \to \infty$, unless $N = 3$ from the beginning.

These two examples are straightforward as the bath-bulk decoupling happens in real space. However, a similar decoupling can happen in energy space (that is, in the basis given by the eigenstates of the Hamiltonian) even if all the sites are connected in real space. In model (d) all the sites are coupled to each other with the same strength $J$; particles are created through a Lindblad bath that acts locally on site 1. We diagonalize Hamiltonian (17) to obtain its eigenoperators $f_i$ and corresponding eigenenergies $\epsilon_i$:

$$f_1 = \frac{1}{\sqrt{3}}(c_1 + c_2 + c_3) \quad \epsilon_1 = -2J,$$
$$f_2 = \frac{1}{\sqrt{2}}(c_1 - c_3) \quad \epsilon_2 = J, \tag{24}$$
$$f_3 = \frac{1}{\sqrt{2}}(c_2 - c_3) \quad \epsilon_3 = J.$$

| Case | Multiplicity of $\lambda_0$ |
|------|------------------------------|
| (b)  | 4 |
| (c)  | 6 |
| (d)  | 2 |
| (e)  | 1 |
| (f)  | 4 |

Table 1: Multiplicity of the zero eigenvalue of the Lindbladian $\mathcal{L}$ in Eq. (13) for the examples considered in Fig. 4.

A double degenerate eigenenergy in the spectrum ($\epsilon_2 = \epsilon_3 = J$) implies a double degenerate subspace, where the basis vectors can be linearly combined so that one of the physical sites has zero weight on one of the two associated eigenstates. This is a consequence of the $\mathbb{Z}_3$ rotational symmetry of the system. From Eq. (24) we see that if the bath is locally connected to site 1 then the eigenoperator $f_3$ is decoupled from it. Combining Eqs. (22) and (24) we obtain the differential equations for the correlation matrix elements $\tilde{C}_{i,i} = \langle f_i^\dagger f_i \rangle = \langle \tilde{n}_i \rangle$ and $\tilde{C}_{i,i+1} = \langle f_i^\dagger f_{i,i+1} \rangle$ in the eigenoperator basis. Then,

$$\langle \dot{\tilde{n}}_3 \rangle := \frac{d}{dt} \langle f_3^\dagger f_3 \rangle = 0. \tag{25}$$

Thus, there is an infinite set of solutions for the stationary state where $\langle \tilde{n}_3 \rangle$ is set only by the initial conditions while $\langle \tilde{n}_1 \rangle = \langle \tilde{n}_2 \rangle = 1$. Note that the degeneracy in the spectrum is broken by a magnetic flux $\phi$ different from 0 or $\pi$, for which $\epsilon_2 \neq \epsilon_3$ and all the system eigenstates have a nonzero weight on each site. Therefore, for $\phi \neq 0, \pi$, the total occupation reaches its maximum $N = 3$ at $t \to \infty$. However, for $\phi = 0$ or $\phi = \pi$, the system is prevented from reaching full occupancy.

Cases (b) and (d) are essentially equivalent, the main difference being that the bath-bulk decoupling happens in real space and energy space, respectively. A similar behavior can be observed in cases (e) and (f) where the three dots are connected to form a short chain with open boundary conditions. When all the hoppings have the same strength $J$, the eigenenergies and eigenoperators are of the form $\epsilon_n = -2J\cos(k_n)$ and $f_n = \mathcal{N} \sum_{i=1}^3 \sin(k_n i) c_i$, respectively, with $k_n = \pi n/4$, $n = 1, 2, 3$ and $\mathcal{N}$ a normalization constant. Explicitly,

$$\begin{aligned}
f_1 &= \frac{1}{2}\left(c_1 + \sqrt{2}c_2 + c_3\right) & \epsilon_1 &= -\sqrt{2}J \\
f_2 &= \frac{1}{\sqrt{2}}\left(c_1 - c_3\right) & \epsilon_2 &= 0 \\
f_3 &= \frac{1}{2}\left(c_1 - \sqrt{2}c_2 + c_3\right) & \epsilon_3 &= \sqrt{2}J.
\end{aligned} \tag{26}$$

While no nodes are allowed on the first and last site of a homogeneous chain, this does not apply for the bulk sites: if the number of sites is odd then the eigenoperators can have a null weight on the bulk sites. In our trimer, the eigenoperator $f_2$ is null on site 2. It follows that, if the local bath acts on site 1 as in model (e) the Lindblad jump operator can populate all three eigenstates of the Hamiltonian and we expect the full system to thermalize (that is, $\lim_{t\to\infty} N = 3$). On the contrary, if the local bath acts on site 2 only, $\langle \dot{\tilde{n}}_2 \rangle = 0$. Thus, the system cannot thermalize, similarly to cases (b), (c), and (d). In Table 1, we report the multiplicity of the zero eigenvalue of $\mathcal{L}$, $\lambda_0$, associated with the steady state. As discussed above, except for case (e), all other cases exhibit a nontrivial manifold of stationary states. In each stationary state, the subsystem connected to the bath is in thermal equilibrium with it, while the rest of the system can in principle be in any state. The multiplicity of $\lambda_0$ minus one corresponds to the

number of degrees of freedom of the decoupled subsystem. Note that the correlation matrix $\mathcal{C}$ contains only a fraction of all the degrees of freedom of the system, the other being stored into the $n > 2$-operator correlators. For this reason, for a complete treatment of the problem, one must solve the full Lindblad equation (13) and investigate the properties of the spectrum of $\mathcal{L}$.

Similar degeneracies for the steady state of the Lindbladian would occur in the Anderson and Aubry-André-Harper chains in the localized phase coupled to the local Lindblad bath acting on the leftmost site if the number of sites in the chain is sent to infinity first, before coupling the system to the bath. The single-particle eigenstates centered on sites infinitely far from the one coupled to the bath would be completely disconnected from it and would not reach thermal equilibrium. However, in finite systems, the decoupling is not perfect, and the degeneracy of the null eigenvalue of $\mathcal{L}$ is not exact. This is because the overlap of single-particle eigenstates on the leftmost site, where the bath acts, is not exactly zero but exponentially small with distance to the localization center.

## 5 Conclusions

We numerically investigated noninteracting disordered chains, specifically the Anderson and Aubry-André-Harper models, coupled to a local Lindblad bath acting on the leftmost site to probe the avalanche instability proposed in the context of many-body localization. By examining the disorder-averaged Lindbladian gap $\Delta$ rescaled by $2^L$ over varying system sizes and disorder strengths, we estimated the avalanche critical disorder strength and uncovered strong finite-size effects. In both models, we observe a characteristic crossover: results for small-to-intermediate finite-size systems indicate ergodicity for increasingly larger disorder strengths as system size increases. However, at larger system sizes, localization appears at progressively smaller disorder values as system size further increases. These finite-size effects are more pronounced in the Anderson model than in the Aubry-André-Harper model, echoing recent observations that quasiperiodic systems are less affected by finite-size and finite-time effects than systems with uncorrelated disorder. Hence, our findings show that, for noninteracting disordered chains, the avalanche critical disorder strength estimated from finite-size numerics within the Lindblad framework may overestimate the avalanche instability threshold in the case of infinitely long chains. This suggests the need for caution when extrapolating finite-size results also for interacting many-body localized systems, as finite-size effects may as well overestimate the critical disorder [96]. We relate the exponential decay of the Lindbladian gap with system size to the bath's limited ability to induce relaxation in larger localized systems. As system size grows, the fraction of eigenstates significantly coupled to the local bath decreases, eventually leading to infinitely long relaxation times for infinitely long chains. Interesting future directions involve the application of the local Lindblad bath framework to other one-dimensional systems, as one-dimensional topological chains with correlated or uncorrelated disorder and/or long-range hopping. In these contexts, nontrivial localization/delocalization transitions occur simultaneously with topological phase transitions [108–111].

## Acknowledgements

**Funding information**  This work received funding from the European Research Council (ERC) under the European Union's Horizon 2020 research and innovation program (Grant Agreement No. 101001902) and the Knut and Alice Wallenberg Foundation (KAW) via the project Dynamic Quantum Matter (2019.0068). A.N. acknowledges funding by the Deutsche Forschungs-

gemeinschaft (DFG, German Research Foundation) under Projektnummer 277101999 - TRR 183 (project C01), under Project No. EG 96/13-1, and under Germany's Excellence Strategy - Cluster of Excellence Matter and Light for Quantum Computing (ML4Q) EXC 2004/1 - 390534769. The computations were enabled by resources provided by the National Academic Infrastructure for Supercomputing in Sweden (NAISS), partially funded by the Swedish Research Council through grant agreement no. 2022-06725.

# A   Details on the Lindbladian gap

## A.1   Numerical precision

As discussed in Sec. 2.2.1, the Lindbladian gap $\Delta$ decreases exponentially with system size as $\kappa^{-L}$, where $\kappa$ grows with disorder strength. For large systems or high disorder, $\Delta$ may become smaller than machine precision, making conventional numerical diagonalization impractical. When $\Delta$ reaches approximately $10^{-14}$ (for double precision), we address this by using Python-FLINT, a Python wrapper for an arbitrary-precision C library by F. Johansson [112]. Although this method is slower and scales less efficiently, it allows analysis of system sizes up to 120 near the avalanche critical disorder.

## A.2   Scaling the Lindbladian gap by $4^L$

Fig. 5 shows the Lindbladian gap $\Delta$ scaled by $4^L$ and averaged over disorder $\langle\log_{10}(\Delta 4^L)\rangle$ as a function of system size and disorder strength for both the bath-coupled Anderson and Aubry-André-Harper models. The $4^L$ scaling has been employed to estimate the avalanche critical disorder in interacting disordered chains [43, 44].

Consistent with Sec. 3, in the Anderson case at small system sizes we observe the presence of an apparent ergodic regime at disorder values that instead show localization for larger system sizes (see, for instance, $W = 0.9, 1.1$). This is indicated by the initial increase and subsequent decrease of the curves for increasing $L$ in panel (a). We estimate the critical disorder strength at finite system size by employing linear interpolation to compute the intersection point between curves of $\langle\log_{10}(\Delta 4^L)\rangle$ as a function of $W$ (or $\lambda$) for close $L$ values (e.g., $L = 6$ and $L = 4$, etc.), as discussed in Sec. 3.1. The finite-size crossing in the inset of (b) yields a critical disorder $W^*(L)$ significantly drifting towards lower values as $L$ increases, similarly to the results obtained with the $2^L$ scaling in Sec. 3.1.

Panels (c) and (d) depict $\langle\log_{10}(\Delta 4^L)\rangle$ as a function of $L$ and $\lambda$ for the Aubry-André-Harper chain coupled to the local Lindblad bath on the leftmost site. By applying the same finite-size crossing analysis as done for the Anderson model, we observe that $\lambda^*(L)$ stabilizes around 1.9 already at system sizes available to us. This shows once again the presence of weaker finite-size effects than in the Anderson case.

## A.3   Spectral gap spread

Fig. 6 shows the standard deviation of the logarithm of the spectral gap $\sigma\left(\log_{10}(\Delta)\right)$ for the two bath-coupled models. In the Anderson case, $\sigma$ is monotonically increasing with disorder strength, whereas in the Aubry-André-Harper model $\sigma$ is maximal near the localization transition $\lambda_c = 1$ of the closed chain and decreases for higher disorder values. A heuristic explanation for these distinct behaviors is as follows. In the Anderson model, as disorder increases the number of possible disorder configurations grows, resulting in a broader spread of the data. In contrast, the correlated disorder in the Aubry-André-Harper model leads to

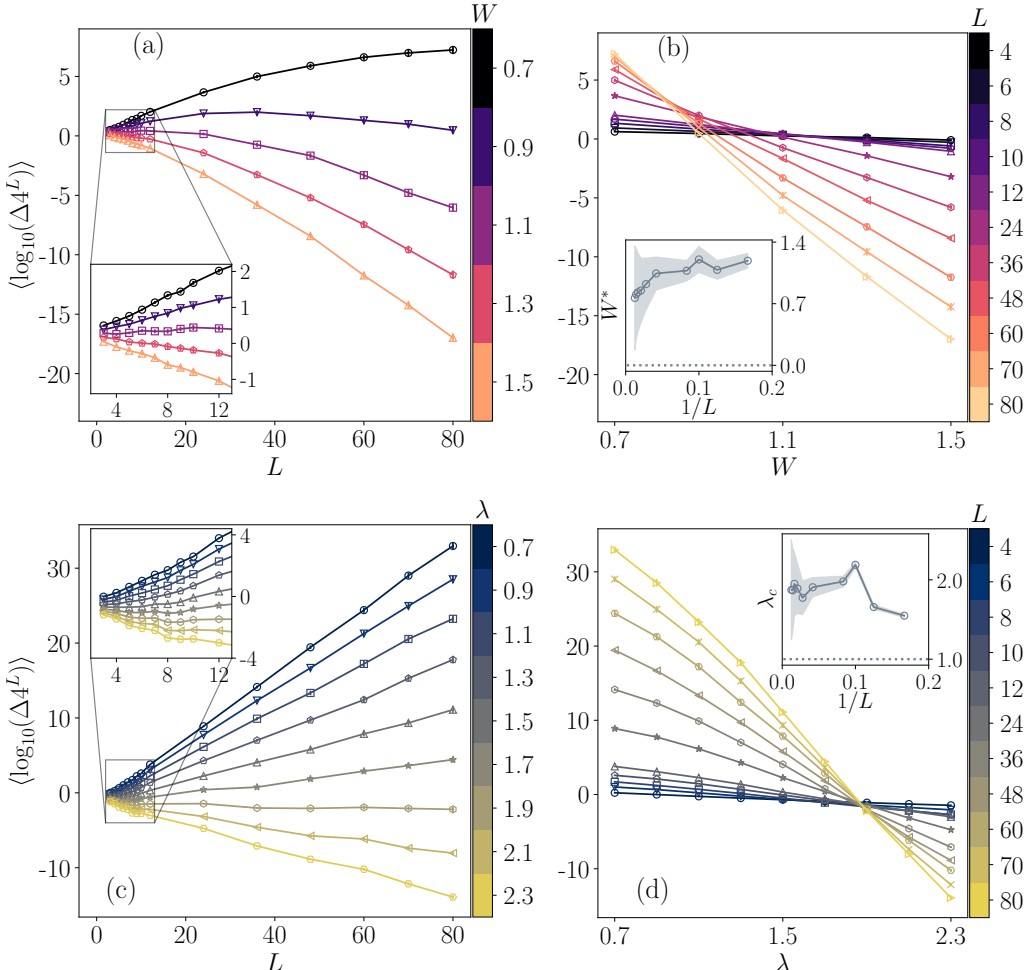

Figure 5: $\langle \log_{10}(\Delta 4^L) \rangle$ for the Anderson [(a)-(b)] and Aubry-André-Harper [(c)-(d)] chain coupled to a local bath acting on the leftmost site as a function of system size $L$ and disorder strength $W$ or $\lambda$. Dotted lines in the insets of (b) and (d) mark the critical disorder strength of the closed chains: $W_c = 0$ for the Anderson chain and $\lambda_c = 1$ for the Aubry-André-Harper chain. We set $J = \gamma = 1$, and average over at least 2000 disorder realizations for each data point. Error bars are 95% bootstrap confidence intervals.

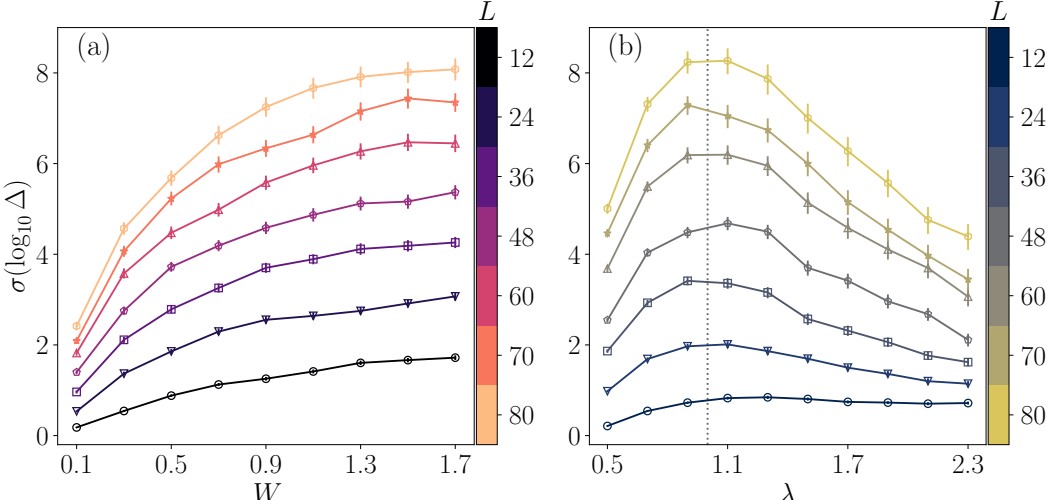

Figure 6: (a) Standard deviation of the logarithm of the spectral gap $\sigma(\log_{10}\Delta)$ as a function of the disorder strength for (a) bath-coupled Anderson model, and (b) bath-coupled Aubry-André-Harper model. (b) The vertical dotted line marks the closed system critical disorder $\lambda_c = 1$. We set $J = \gamma = 1$. Error bars are 95% bootstrap confidence intervals.

increased similarity and correlations among different disorder configurations as disorder becomes larger than $\lambda_c = 1$. Consequently, the spread of the data diminishes.

## B Results for small system sizes in the presence of dephasing

Recent works, as for instance Refs. 89–92, have demonstrated that bath-induced dephasing operators acting locally along the entire chain can trigger anomalous transport even in localized chains. A dephasing operator on the $i$-th site $L_{3,i} = 2c_i^\dagger c_i$ might annihilate a particle in one eigenstate and recreate it in another eigenstate that overlaps with the first on site $i$. Effective transport across the $i$-th site occurs if the localization centers of the two involved eigenstates are on opposite sides with respect to site $i$.

In our framework, the bath acts only on the leftmost site of the chain, creating and annihilating particles ($L_1 = c_1^\dagger$, $L_2 = c_1$; see Eqs. (14)-(15)). The addition of a dephasing term acting on the leftmost site $L_{3,1} = 2c_1^\dagger c_1$ would only affect eigenstates with overlap on site 1. Its effect would be similar to that of $L_1$ and $L_2$ Lindblad jump operators together. Hence, a boundary-dephasing term only affects the results discussed in the main text quantitatively but not qualitatively. As the system size or disorder strength increases, the number of eigenfunctions not coupled with the bath increases, reducing the impact of such a dephasing term.

In Fig. 7 we compare results for the rescaled Lindblad spectral gap $\langle \log_{10}(\Delta 2^L) \rangle$ as a function of $L$ and $W$ in the presence and in the absence of the boundary dephasing jump operator $L_{3,1} = 2c_1^\dagger c_1$. As expected, the two behaviors are qualitatively similar, with the mismatch between the two cases becoming less significant for increasing $L$ and $W$.

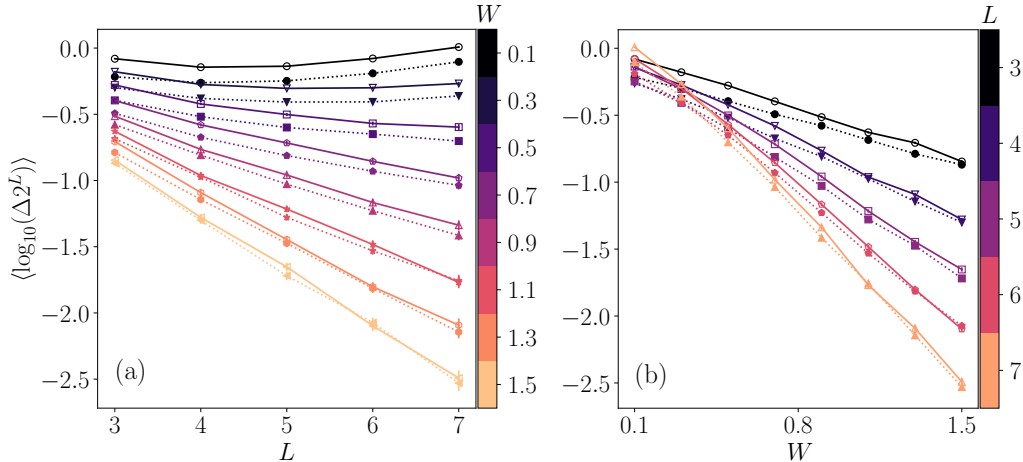

Figure 7: Lindbladian gap $\Delta$ scaled by $2^L$ and averaged over disorder for the Anderson model as a function of (a) $L$, and (b) $W$. Empty symbols connected by solid lines reproduce the results in the main text in the absence of dephasing. Filled symbols connected by dotted lines are obtained in the presence of dephasing on the leftmost site of the chain, $L_{3,1} = 2c_1^\dagger c_1$. We set $J = \gamma = 1$, and average over at least 2000 disorder realizations for each data point. Error bars are 95% bootstrap confidence intervals.

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
