# Peer review of "Numerical Study of Disordered Noninteracting Chains Coupled to a Local Lindblad Bath"

_SciPost Physics_

## Round 1 · Referee Report · Anonymous (Referee 1) · 2025-1-14

Report

Warnings issued while processing user-supplied markup:

  • Inconsistency: plain/Markdown and reStructuredText syntaxes are mixed. Markdown will be used.
    Add "#coerce:reST" or "#coerce:plain" as the first line of your text to force reStructuredText or no markup.
    You may also contact the helpdesk if the formatting is incorrect and you are unable to edit your text.

This paper studies whether localization persists in an noninteracting disordered spin chain whose edge is coupled to local baths, which may trigger avalanche instability and lead to thermalization. The key insight is that the scaling behavior of the Lindbladian gap could be used as a diagnostic of ergodicity/localization. In particular, this paper numerically computed the Lindbladian gap for the Anderson and Anbry-Andre-Harper chain (with the left boundary coupled to spin raising and lowering operators) for various system sizes and disorder strengths. This paper examines the spectral gap rescaled by $2^L$ ($L$ is the system size) and finds qualitatively different scaling behaviors for disorder strengths larger and smaller than a critical threshold. This finding is corroborated by the behavior of percent of eigenstates with non-vanishing weight on the first site for different disorder strengths. Finite size effects were also explored. This paper reports strong finite size effects for intermediate system sizes and shows that the critical disorder threshold approaches the expected value from above for longer chains.

I have a few major comments and questions: 1. The motivation/main result of this work is not entirely clear. As the authors have also noted, thermal avalanches can only arise in interacting chains. So what exactly does simulating non-interacting chains do? Surely, much larger system sizes will be accessible numerically than the interacting case, but how does the extrapolated disorder threshold relate to the "actual" disorder threshold in interacting chains? The paper only suggests that the extrapolated value is larger than the actual critical disorder in closed chains. Can we say more about this relation? For example, can we claim the extrapolated value approaches the actual value (from above) in the thermodynamic limit? Or is it just an upper bound (if so, why?) The physics isn't really clear. Perhaps the authors could shed some light on this.

  1. The mapping between a dissipative spin chain and a non-Hermitian two-leg ladder has been rather well-established (and more generally between a dissipative lattice model and its non-hermitian bilayer counterpart; see e.g. PRB 99 174303, PRB 109,085115). Since the system is a non-interacting chain of spin-halves, would it be possible to map it to a non-Hermitian problem quadratic in fermion operators and even solve it exactly? While this is a numerical study, writing down the problem in the fermion language could add more completeness to the work.

  2. The motivation of Section 4 is not clear. The main point is that when there is decoupling in either real or energy space, certain degrees of freedom get a divergent lifetime/ vanishing Lindbladian gap, and this mechanism applies to the dissipative disordered chains at hand (some states can be completely decoupled from the first site and therefore the bath). However, this argument is rather straight-forward, well-known, and intuitive. A few paragraphs might have done the job of establishing this connection. Could the authors motivate/argue why a study of the toy model is needed/ should be kept in the paper?

  3. Figures: i. using legends instead of colormaps might have been sufficient for all the plots. ii. Figure 2(c) and similarly 5(c): the authors should consider using a plot marker other than the stars (lowest curve in 2c, $\lambda = 1.7$). The stars cover the error bars almost completely. iii. Figure 3: using log-scale for the y axis might work better. Is there a particular reason why a linear scale was chosen instead? iv. Figure 6: it's not clear why there are still error bars. Is the plotted quantity already the standard deviation across all disorder realizations?

In addition, I have a few minor comments/suggestions: 1. In the second paragraph of Sec 2.2.1, is there a reason to assume/claim the Lindbladian gap scales as the thermalization rate?

  1. On page 4 under Eq. 6: is there a way to quantify "weak" vs. "strong" disorder? (a comparison of W against some other energy scale?)

  2. The authors might consider going beyond simply citing third quantization and including more details on how the numerics were performed.

  3. In computing the overlap of single-particle eigenstates with the first site, the paper chooses $10^{-14}$ as the error threshold. Why was the machine precision used? Naively, the threshold for determining whether there is significant overlap should depend on the system size as $1/L$. Would the claims in the paper remain valid with a system-size dependent threshold? Or might $10^{-14}$ suffice? Perhaps the authors can help clarify this point.

  4. Typos: i. Last paragraph of Sec 2.1.1: The sentence "If in a finite chain..." needs a bit of rewording for better clarity. ii. Right above Eq. 8, the golden ratio should be $(\sqrt{5} + 1 )/2$ instead of $-$. iii. At bottom of page 9: an extra space between $W$ and $(\lambda)$. iv. Eq. 17: consider using a dummy index other than $i$. There is already an imaginary $i$ in the phase term $e^{i \phi/3}$.

Overall, I think the paper studies an interesting question, is largely valid, and is honestly and cleanly written. However, a few major things aren't exactly clear, and the authors are politely recommended to address the comments above.

Requested changes

See comments above.

Recommendation

Ask for minor revision

  • validity: good
  • significance: ok
  • originality: good
  • clarity: high
  • formatting: good
  • grammar: excellent

Author:  Claudia Artiaco  on 2025-10-09  [id 5909]

(in reply to Report 1 on 2025-01-14)
Category:
answer to question

\(\textbf{Referee:}\) This paper studies whether localization persists in an noninteracting disordered spin chain whose edge is coupled to local baths, which may trigger avalanche instability and lead to thermalization. The key insight is that the scaling behavior of the Lindbladian gap could be used as a diagnostic of ergodicity/localization. In particular, this paper numerically computed the Lindbladian gap for the Anderson and Anbry-Andre-Harper chain (with the left boundary coupled to spin raising and lowering operators) for various system sizes and disorder strengths. This paper examines the spectral gap rescaled by \( 2^L\)(\(L\) is the system size) and finds qualitatively different scaling behaviors for disorder strengths larger and smaller than a critical threshold. This finding is corroborated by the behavior of percent of eigenstates with non-vanishing weight on the first site for different disorder strengths. Finite size effects were also explored. This paper reports strong finite size effects for intermediate system sizes and shows that the critical disorder threshold approaches the expected value from above for longer chains.

\(\textbf{Authors:}\) We thank the referee for the succinct summary of our work.

\(\textbf{Referee:}\) I have a few major comments and questions: 1. The motivation/main result of this work is not entirely clear. As the authors have also noted, thermal avalanches can only arise in interacting chains. So what exactly does simulating non-interacting chains do? Surely, much larger system sizes will be accessible numerically than the interacting case, but how does the extrapolated disorder threshold relate to the ``actual" disorder threshold in interacting chains?

\(\textbf{Authors:}\) We thank the referee for the question. Indeed, the propagation of thermal avalanches in closed systems relies on interactions and do not occur in noninteracting systems. However, the local Lindblad bath framework assumes from the outset that the avalanche mechanism is active, even if not present in the microscopic Hamiltonian dynamics. Within this formalism, we use the Lindbladian gap to estimate the slowest relaxation rate of the system coupled to the local bath, and we interpret its scaling with system size as a measure of the stability of the localized phase against thermal avalanches. Our Lindbladian setting can be considered a toy model, derived within a series of assumptions that we thoroughly discuss in Sec.2, for the stability against thermal avalanches of interacting disordered systems. Specifically, in an interacting system ergodic bubbles would be naturally present due to regions where disorder is weak. As in our noninteracting chain ergodic bubbles do not naturally arise, we do insert them on purpose by the local bath on one boundary. Then, the noninteracting localized chain can be seen as a minimal model for the interacting chain deep in the localized regime.

One of the aims of our work is to critically examine the local Lindblad bath framework used for probing the avalanche instability in interacting disordered chains in several recent studies (e.g., Refs.[43,44,73,74]) by applying it to noninteracting disordered chains, for which much larger system sizes can be accessed. This allows us to investigate whether the scaling of the Lindbladian gap with system size indeed reflects the known localization properties of the model. We find that, even in this simplified setting, strong finite-size effects up to quite large system sizes (\(\sim \mathcal{O}(100)\)) can lead to an overestimation of the critical disorder when using the Lindblad bath protocol. This suggests that similar effects might influence extrapolations in the interacting case, where numerical access is limited to much smaller systems. Therefore, our study serves as a benchmark to assess the reliability and limitations of using Lindblad-based diagnostics to infer the avalanche instability threshold and emphasizes the need for caution when interpreting finite-size data in many-body localized systems.

Given the assumptions made, namely, first the assumption that the actual relaxation rate of the system corresponds to the inverse of the Lindbladian gap and second restricting the study to noninteracting localized chains, we cannot make any solid statement about the relation between the extrapolated avalanche critical disorder and the critical disorder in interacting disordered chains. This is why we do not make any statement in the manuscript. However, there is no reason to think that the strong finite-size effects we analyzed for the noninteracting case are not present in the interacting case. Therefore, we can argue that using the same local Lindblad bath framework for investigating the stability of disordered interacting chains against thermal avalanches suffers from similar strong finite-size effects. Thus, results obtained for small system sizes, which are the only ones accessible for interacting disordered chains, should be considered with caution as the behavior might drastically change at larger system sizes inaccessible in current numerics.

\(\textbf{Referee:}\) The paper only suggests that the extrapolated value is larger than the actual critical disorder in closed chains. Can we say more about this relation? For example, can we claim the extrapolated value approaches the actual value (from above) in the thermodynamic limit? Or is it just an upper bound (if so, why?) The physics isn't really clear. Perhaps the authors could shed some light on this.

\(\textbf{Authors:}\) Our numerical results indicate that the avalanche critical disorder for the local Lindblad bath framework at finite system size, \(W^*(L)\) and \(\lambda^*(L)\), is higher than the asymptotic value of the avalanche critical disorder extrapolated for infinite system size, \(W^*_\infty\) and \(\lambda^*_\infty\). Moreover, \(W^*_\infty\) and \(\lambda^*_\infty\) are larger than the actual critical disorder in the closed chains \(W_c\) and \(\lambda_c\). The latter result is well expected as our local Lindblad framework assumes from the outset that the avalanche takes place and evaluates it from the scaling of the Lindbladian gap with system size compared to \(2^{-L}\). As we write at the end of Sec.2.2.1 of the manuscript, the scaling of \(\Delta\) with system size in localized systems is known to be exponential, i.e., \(\Delta \propto e^{-L/\zeta}\) with \(\zeta\) the decay length of the Lindbladian gap, which depends on the disorder strength. This scaling is discussed in Ref.[82] and further analyzed in Ref.[83], which we refer to in the manuscript. This implies that our framework always sets the avalanche threshold to be at a disorder strength larger than the critical disorder of the closed chains, namely when \(\zeta \sim \frac{1}{\log(2)}\). As \(\zeta\) is infinite in the delocalized phase and takes finite values in the localized phase, this implies that \(W^*(L)\) and \(\lambda^*(L)\) must be larger than the critical disorder values in the closed chains.

\(\textbf{Referee:}\) 1) The mapping between a dissipative spin chain and a non-Hermitian two-leg ladder has been rather well-established (and more generally between a dissipative lattice model and its non-hermitian bilayer counterpart; see e.g. PRB 99 174303, PRB 109,085115). Since the system is a non-interacting chain of spin-halves, would it be possible to map it to a non-Hermitian problem quadratic in fermion operators and even solve it exactly? While this is a numerical study, writing down the problem in the fermion language could add more completeness to the work.

\(\textbf{Authors:}\) We thank the referee for suggesting to us some interesting works that we have now appropriately cited in the paper (as Ref.[99: PRB 99 174303] and Ref.[100: PRB 109, 085115]). Although the proposed technique is effective for solving the problem analytically under suitable conditions, for example unitary jump operators as in [100], or by means of controlled approximations (no click term limit or non-Hermitian perturbation theory), it is not helpful to find an exact solution for the disordered case, in which we are interested. The non-Hermitian Hamiltonian should then be solved by means of numerical approaches not simpler than the third quantization we use to compute \(\Delta\), which exploits as well the quadratic nature of the problem, or by resorting to the two-body correlation matrix, as done in Sec.4 in our manuscript. Considering the referee's comment, at the beginning of page 9 of the paper we added “It is worth noting that, in the absence of disorder, analytical treatments of the problem are possible mapping the one-dimensional chain to a non-Hermitian ladder (see Refs.[99,100]).”

\(\textbf{Referee:}\) 2) The motivation of Section 4 is not clear. The main point is that when there is decoupling in either real or energy space, certain degrees of freedom get a divergent lifetime/vanishing Lindbladian gap, and this mechanism applies to the dissipative disordered chains at hand (some states can be completely decoupled from the first site and therefore the bath). However, this argument is rather straight-forward, well-known, and intuitive. A few paragraphs might have done the job of establishing this connection. Could the authors motivate/argue why a study of the toy model is needed/ should be kept in the paper?

\(\textbf{Authors:}\) We thank the referee for the question. As the referee 2 writes, Sec.4 provides a simple toy model to understand the origin of the effective decoupling of the system from the bath and for the exponential decay of the Lindbladian gap with system size. Even though we agree that the results are straightforward, we do not believe that they stop the flow of the paper or confuse the reader as they are placed after the thorough discussion of the main results. Rather, we believe that this section, thanks to its simple content, helps convey the main message of our manuscript and the results, as also acknowledged by referee 2. Therefore, we prefer to keep this section in the main manuscript rather than move it to the Appendices.

\(\textbf{Referee:}\) 3) Figures: i. using legends instead of colormaps might have been sufficient for all the plots. ii. Figure 2(c) and similarly 5(c): the authors should consider using a plot marker other than the stars (lowest curve in 2c, \(\lambda=1.7\)). The stars cover the error bars almost completely. iii. Figure 3: using log-scale for the y axis might work better. Is there a particular reason why a linear scale was chosen instead? iv. Figure 6: it's not clear why there are still error bars. Is the plotted quantity already the standard deviation across all disorder realizations?

\(\textbf{Authors:}\) We thank the referee for the helpful comments. i. We appreciate the referee's suggestion. However, we prefer to keep the colorbars instead of legends as we think they improve the readability of our plots. ii. We thank the referee for the suggestion. We updated the plots in Figures 2, 3, 5, 6, 7 in the manuscript by substituting the star and other symbols with more readable ones. We also increased the marker size to enhance the visibility of the error bars. iii. We thank the referee for the suggestion. However, we verified that the linear scale for the y-axis provides a better readability of the plots and clearly conveys the features discussed in the text, namely, that as system size and disorder increase the percentage of single-particle Hamiltonian eigenstates with nonvanishing overlap on the leftmost site of the chain rapidly decreases. iv. We thank the referee for pointing this out. In Fig.6 the points themselves are the sample standard deviations of \(\log_{10}(\Delta)\) over all disorder realizations at each parameter set, and the vertical error bars are the 95% confidence intervals on those standard‐deviation estimates, obtained via bootstrap. Concretely, for each disorder ensemble we resample (with replacement) the same number of \(\log_{10}(\Delta)\) values, compute the standard deviation in each resample, and take the 2.5% and 97.5% quantiles of that resampled distribution as the lower and upper limits of the error bar. This is a standard method for estimating the uncertainty (see, e.g., B. Efron and R. Tibshirani, An Introduction to the Bootstrap (Chapman & Hall, 1993) or the scipy.stats documentation on the bootstrap function at https://docs.scipy.org/doc/scipy/reference/generated/scipy.stats.bootstrap.html).

\(\textbf{Referee:}\) 4) In addition, I have a few minor comments/suggestions: 1. In the second paragraph of Sec 2.2.1, is there a reason to assume/claim the Lindbladian gap scales as the thermalization rate?

\(\textbf{Authors:}\) We thank the referee for raising this point. In Lindblad dynamics, the spectrum of the Lindbladian superoperator governs the relaxation of the system toward its steady state. The nonzero eigenvalue with the largest real part smaller than zero, \(\lambda_1\), determines the slowest exponential decay mode. This can be understood by considering the solution to the Lindblad equation (Eq.(16) in the manuscript). Indeed, \(\frac{1}{-\text{Re}(\lambda_1)} = \frac{1}{\Delta}\) is the longest timescale of the system; for \(t \gtrsim \frac{1}{\Delta}\), \(\rho(t)\) has a negligible overlap with all right eigenmatrices \(R_i\) different from \(R_{0,m}\), where \(R_{0,m}\) are the steady states associated with \(\lambda_0=0\). Thus, \(\frac{1}{\Delta}\) provides a natural estimate of the longest relaxation time (also called the mixing or thermalization time). This relationship is well established in the theory of quantum Markov processes (see Ref.[80] in the manuscript). In particular, in the context of localization physics, the same assumption about the relation between the Lindbladian gap and the thermalization rate has been taken in previous works studying thermal avalanches (see Refs.[43,44,73,74] in the manuscript). Considering the referee's question, we added a discussion about this after Eq.(16).

\(\textbf{Referee:}\) 5) On page 4 under Eq. 6: is there a way to quantify "weak" vs. "strong" disorder? (a comparison of W against some other energy scale?)

\(\textbf{Authors:}\) We thank the referee for the question. In the context of localization in noninteracting disordered systems, weak and strong disorder are usually defined by comparing the disorder strength \(W\) to the hopping amplitude \(J\), which sets the characteristic energy scale of the clean system. We have modified the manuscript accordingly to clarify this point.

\(\textbf{Referee:}\) 6) The authors might consider going beyond simply citing third quantization and including more details on how the numerics were performed.

\(\textbf{Authors:}\) We thank the referee for the suggestion. We have added a short summary with some more details on third quantization and, in particular, on how to compute \(\Delta\) within third quantization at the end of Sec.2.2.2. We furthermore explicitly provide the relevant reference containing all the details, namely Ref.[82].

\(\textbf{Referee:}\) 7) In computing the overlap of single-particle eigenstates with the first site, the paper chooses \(10^{-14}\) as the error threshold. Why was the machine precision used? Naively, the threshold for determining whether there is significant overlap should depend on the system size as \(1/L\). Would the claims in the paper remain valid with a system-size dependent threshold? Or might \(10^{-14}\) suffice? Perhaps the authors can help clarify this point.

\(\textbf{Authors:}\) We thank the referee for this observation. In Sec.3.2 we are interested in showing the percentage of states with a nonvanishing weight on the first site, \(p\), as a function of disorder strength and system size. For this reason we fixed the machine precision threshold \(10^{-14}\) as an overlap indicator. A threshold equal to \(1/L\) would select states whose overlap with the first site is still large enough to allow their thermalization. Clearly, in the thermodynamic limit, \(L \rightarrow \infty\), the two definitions would provide similar results but, at finite size, the \(1/L\) threshold would overestimate \(p\) (even if it would predict a qualitatively similar result).

\(\textbf{Referee:}\) 8) Typos: i. Last paragraph of Sec 2.1.1: The sentence "If in a finite chain..." needs a bit of rewording for better clarity. ii. Right above Eq. 8, the golden ratio should be \((\sqrt{5}+1)/2\) instead of \(-\). iii. At bottom of page 9: an extra space between \(W\) and \((\lambda)\). iv. Eq. 17: consider using a dummy index other than \(i\). There is already an imaginary \(i\) in the phase term \(e^{i\phi/3}\).

\(\textbf{Authors:}\) We thank the referee for the helpful comments. i. We rephrased the last paragraph of Sec.2.1.1 to enhance clarity. ii. We corrected the expression for the golden ratio. iii. We did not find any extra space between \(W\) and \((\lambda)\). The appearance of additional spacing is likely due to LaTeX formatting and does not reflect an actual typo. iv. We appreciate the referee’s suggestion. However, since the index \(i\) is consistently used throughout the manuscript to denote the physical site index, we believe there is no risk of confusion and prefer to retain the same notation in Eq.(17) for consistency.

Anonymous on 2025-10-09  [id 5910]

(in reply to Claudia Artiaco on 2025-10-09 [id 5909])

I believe the authors have addressed all of my comments. I concur that this paper is suitable for publication in SciPost Phys Core.

---

## Round 1 · Referee Report · Anonymous (Referee 2) · 2025-2-28

Report

The manuscript by Berger and collaborators studies the physics of Anderson localisation in one dimensional chains (with random or quasiperiodic disorder) in presence of a dissipative coupling to a bath at one edge of the chain. The motivation for this work comes from the Many-Body-Localization transition and its possible instability due to thermal avalanches.

The Authors take advantage of the non-interacting nature of the problem and use the formalism of third quantization to solve their models and compute the Lindbladian gap. The scaling of this quantity with system size, for different disorder strengths, allows to obtain information on the stability of the localised phase. The main result of this work is that the Lindbladian gap, related to the inverse relaxation time, scales to zero exponentially for large chains. In addition, the Authors point out that for both models under scrutiny (and in particular for the case of quenched random disorder) the finite size effects are strong enough to prevent obtaining a conclusive picture. Nevertheless, the numerical evidence suggests an exponential decay the critical disorder for the avalanche instability is overestimated. The Authors conclude their work by providing a simple toy model to understand the origin of the effective decoupling of the bath from the system, at the origin of the exponential scaling of the gap.

I think the content of this manuscript is interesting and worth to be published in a suitable journal. However the manuscript does not contain in my opinion sufficient material and results to justify publication in SciPost Physics, based on its acceptance criteria which require either a major advancement/breakthrough or opening of a new research direction. In its current form I would be happy to recommend the paper for Scipost Phys. Core.

Recommendation

Accept in alternative Journal (see Report)

  • validity: high
  • significance: good
  • originality: good
  • clarity: ok
  • formatting: perfect
  • grammar: perfect

Author:  Claudia Artiaco  on 2025-10-09  [id 5908]

(in reply to Report 2 on 2025-02-28)
Category:
answer to question

We thank the referee for the careful reading of our manuscript, for the succinct summary of its main results, and for the positive evaluation of its content and analyses. We indeed believe our work provides an important and thorough evaluation of the applicability of the local Lindblad bath framework to assess the stability of localized systems against thermal avalanches, showing that numerical results and finite size effects for these phenomena should be carefully evaluated.

We appreciate the referee's observation regarding the scope and criteria of SciPost Physics. While we believe our findings are of broad interest to the localization community, we concur that the manuscript aligns well with the aims of SciPost Physics Core. Therefore, we kindly ask the Editor to consider our manuscript for publication in SciPost Physics Core and would be pleased to proceed with publication in that venue.

---

## Editorial Decision

resubmitted